# Estimation of Average Grain Size from Microstructure Image Using a Convolutional Neural Network

**DOI:** 10.3390/ma15196954

**Published:** 2022-10-07

**Authors:** Jun-Ho Jung, Seok-Jae Lee, Hee-Soo Kim

**Affiliations:** 1Department of Advanced Materials Engineering, Chosun University, 309 Pilmun-daero, Dong-gu, Gwangju 61452, Korea; 2Division of Advanced Materials Engineering, Jeonbuk National University, 567 Baekje-daero, Deokjin-gu, Jeonju 54896, Korea

**Keywords:** machine learning, convolutional neural network, image recognition, microstructure, grain size

## Abstract

In this study, the average grain size was evaluated from a microstructure image using a convolutional neural network. Since the grain size in a microstructure image can be directly measured and verified in the original image, unlike the chemical composition or mechanical properties of material, it is more appropriate to validate the training results quantitatively. An analysis of microstructure images, such as grain size, can be performed manually or using image analysis software; however, it is expected that the analysis would be simpler and faster with machine learning. Microstructure images were created using a phase-field simulation, and machine learning was carried out with a convolutional neural network model. The relationship between the microstructure image and the average grain size was not judged by classification, as the goal was to have different results for each microstructure using regression. The results showed high accuracy within the training range. The average grain sizes of experimental images with explicit grain boundary were well estimated by the network. The mid-layer image was analyzed to examine how the network understood the input microstructure image. The network seemed to recognize the curvatures of the grain boundaries and estimate the average grain size from these curvatures.

## 1. Introduction

The mechanical properties of materials are directly related to their microstructure, and material developers are trying to achieve targeted mechanical properties by observing and controlling the microstructure during the development of new materials. In particular, it is well known that the average grain size is strongly related to the strength, hardness, elongation, and fatigue properties of materials [1,2,3]. Many studies have been conducted on refining grains and inhibiting grain growth by precipitating a second phase [4,5]. The most fundamentally measured characteristic when observing a microstructure is the grain size. Since it is difficult to measure grain volume directly from a three-dimensional grain, the grain size was analyzed in the form of an area or radius from a two-dimensional cross-sectional image. The simplest way to measure the average grain size is to draw an arbitrary straight line and divide the length of the line by the number of grains that intersect it [6]. Although this method is simple, it requires a substantial amount of time and effort, and the subjectivity of the observer may influence the results. After computers were commercialized, the grain size could be measured more easily using graphic software, but even then, the subjectivity of the user cannot be completely excluded. Automation of qualitative microstructure analysis using a computer is expected to be possible using machine learning, which has recently become popular, and research on this has been conducted [7,8,9].

Quantitative microstructure analysis has also been used to interpret alloy components [10], material properties [11,12], and microstructural features [13,14] based on microstructure images using machine learning and computer vision. From the point of view of machine learning for image recognition, the prediction of alloy components or material properties and the prediction of average grain size look similar to each other, but there is one major difference. The alloy composition and material properties are sampled from different parts of a product, resulting in an average alloy composition for the entire product. When observing the microstructure, only a few specimens are obtained and only the local area of the product is observed. It is difficult to conclude that the microstructure images obtained from various sites have an average alloy composition owing to the macrosegregation and microsegregation of materials. However, the average grain size may have different values for each part of a product, but accurate results can always be obtained within a microstructure image. Therefore, to clearly determine whether machine learning can quantitatively evaluate microstructure images, an evaluation of the average grain size may be more appropriate than a prediction of alloy composition and material properties. There has been research on grain size measurement by edge detection of grain boundaries using digital image processing [15,16]. Moreover, grain size prediction has been attempted using image classification based on a convolutional neural network (CNN) [17,18].

In this study, a simple CNN was constructed, and the average grain size was estimated from microstructure images. It is very difficult to secure enough actual microstructure images and to measure the sizes of grains in each image for machine learning. Grain structure images generated using the phase-field model were used as training data in this study. The simulation generated many images that were sufficient for network training. Because an image of a grain structure can be expressed in different ways depending on the chemical etching and electron microscopy images, the possibility of recognizing various forms of grain structure presentations was also considered.

## 2. Model Description

### 2.1. Preparation of Microstructure Images

The microstructure images used in this study were generated using a phase-field model of two-dimensional normal grain growth. Since a description of the phase-field model is beyond the scope of this study, refer to References [19,20] for further details. Each image had a size of 512 × 512 pixels and RGB information. We considered two types of simulated microstructure images, that is, GB and CL-types, as shown in Figure 1a,b. In the simulated GB-type images, the grain matrix was white, and the grain boundaries were black, similar to a chemically etched specimen. The simulated CL-type images have colorful grains without distinct grain boundary areas, similar to electron backscatter diffraction (EBSD) micrographs. Note that the color of each grain differs from the colors of other grains. Network training in the present study was performed using only GB-type images. CL-type images were used as additional validation data for determining how the network recognizes microstructure images expressed in other forms.

Since the images were generated by computational simulation, there were no scale bars on them. The size of a grain was defined as the number of pixels in the grain area of a 512 × 512-pixel image. Also, grains which had been cut off by the boundary of the image were treated as one grain. GB was used for the training and testing data, and the total number of datasets was 4000. The average grain size ranged from approximately 1200 to 2400 pixels. The number of grains ranged from 110 to 220. Since it is not difficult to create a microstructure with new grain structures using computational simulations, no special data augmentation technique was used. In normal grain growth, the grain area is linearly proportional to time [21]. Using these characteristics, it was possible to obtain the desired grain size at a certain time step during the simulation. Using this method, the average grain size is evenly distributed within the training range. The data for additional validation were generated, which were not included in the training or test datasets described above.

To apply the network trained above to actual situations, some experimental grain structures were collected across the internet, as shown in Figure 1c,d. The images were cropped to squares excluding scale bars and were resized to 512 × 512-pixel images. The GB-type images were taken from optical microscopy, and the CL-type images were EBSD micrographs. The number of grains in each image was counted, and the total number of pixels was divided by it to manually estimate the average grain size.

### 2.2. Convolutional Neural Network

CNN [22] is a deep learning method for image processing and has a structure similar to an artificial neural network [23]. A schematic structure of the CNN used in this study is shown in Figure 2. The overall structure of a CNN consists of an input layer, hidden layers, and an output layer. Since the purpose of this study was to regress the average grain size, a microstructure image was entered into the input layer as a variable. The initial image was a 512 × 512 × 3 RGB color image and was reduced to a 256 × 256 × 3 color image immediately after input to reduce the use of computer resources. The output layer yielded a predicted average grain size.

The hidden layers consist of CP layers that combine convolution, activation functions, pooling operations, and a fully coupled layer (FC). The convolution layer in a CP layer has multiple channels, and image information is stored in each channel; the current and previous layers are connected by convolution with filters, which are small images. The shapes of the filter images are determined during the network training. The image size is reduced during the convolution process. Padding and stride are applied additionally to control the image reduction during the convolution operation. After the convolution operation, the data are connected to an activation function and a pooling operation that further reduces the image. For details on each operation, refer to references [24,25]. As shown in Figure 2, four CP layers were connected in the network. When an image is sufficiently reduced after passing through CP4, it undergoes a flattening process and is connected to the FC layer, which is a general artificial neural network. Between the flattened and FC layers, a 50% dropout was applied to minimize overfitting. Finally, the FC layer was connected to the output layer to yield the regressed result.

For the activation functions, the rectified linear unit (ReLU) function was applied to the CP and FC layers. ReLU is simpler than a sigmoid function and results in faster training. No activation function was used in the output layer, since the regressed results do not require an activation function for classification. In this study, the number of hidden layers, channels, and nodes was determined by trial and error with the goal of obtaining the minimum loss. Table 1 summarizes the operating conditions of the CNN used in this study. Training proceeds in units of epochs, and one epoch is a training step trained once using all input data. The mean squared error (MSE) and Adam [26] were used as the loss and optimization methods, respectively. CNN was implemented using Python [27] and Keras [28].

## 3. Results and Discussion

### 3.1. Preliminary Analysis of Microstructure

A basic analysis of the grain structure generated by the phase-field model is helpful in interpreting the results of machine learning in subsequent analysis. The average grain size can be measured in two ways using the computational simulation results shown in Figure 3a. The most intuitive and easiest way to determine the average grain size (*A*) is to divide the number of grains in the total image area (*A*_total_) by the number (*n*) of grains in the entire image:(1)$$A=\frac{A_{\mathrm{total}}}{n}$$

The predicted results according to Equation (1) for the 4000 image data points used for training and testing of the CNN are shown in Figure 3a. The predicted values for the number of grains and the average grain size were in perfect agreement with the actual results. This is expected, as Equation (1) was used to define the average grain size.

Since the width of the grain boundary is fixed in the phase-field model [20], the grain boundary area, which appears black in a GB-type image, may be closely related to the grain size. Assuming that the shape of the grains is circular, the ratio (*G*) of the grain boundary area to the average grain area is expressed by the following equation:(2)$$A=\frac{\pi \xi^{2}}{{\left(1-\sqrt{1-G}\right)}^{2}}$$
where ξ is the half width of the grain boundary. Based on Equation (2), the relationship between *G* and *A* was analyzed using 4000 images and is shown in Figure 3b. Although a significant error occurs compared with Figure 3a, which shows the result predicted by the number of grains, it is clear that the average grain size can be predicted by the ratio of the grain boundary areas.

To evaluate the average grain size using machine learning, the network acts like a black box that does not provide the user with a basis for prediction. Comparing the analysis results of Figure 2 and the results of machine learning, it is possible to estimate how the average grain size is predicted by CNN.

### 3.2. Accuracy of Machine Learning

Among the 4000 GB-type images generated with the phase-field model, 3000 randomly chosen images (75%) were used to train the CNN, and the 1000 remaining images (25%) were used to test the network. The change in loss during training for 100,000 epochs is shown in Figure 4. The blue and red lines indicate the results of the training and test data, respectively. The smaller the mean square loss, the better the training performed. As shown in the figure, the loss for the training data decreased rapidly at the beginning of training, and the slope decreased as the training step increased. The loss for the test data decreased in large amplitudes, the overall trend was similar to that of the training data, and overfitting did not occur. After 80,000 epochs, a high-amplitude noise appeared in the training and test data. Therefore, CNN training was performed for up to 80,000 epochs.

A graph comparing the training and test data with the measured data is presented in Figure 5a. The *X*-axis represents the average grain size measured in the computational simulation, which generates the grain structures. The *Y*-axis represents the results predicted by CNN. For the training data, the slope was >0.99, and *R*^2^ was 0.999, indicating an almost perfect agreement with the measured values. This value is almost identical to the value predicted by the number of grains determined using Equation (1). The test data showed a slope of 0.95 and *R*^2^ of 0.979. In terms of the slope and *R*^2^, the accuracy was sufficiently high to be useful, but there was a significant deviation compared with the training data. A graph showing the ratio of the average grain sizes (*A*_predicted_) predicted by the CNN to the measured results (*A*_measured_) is shown in Figure 5b. For the training data, the predicted results were very accurate, with a result of close to 1 for all data. For the test data, the average accuracy was close to 1, and significant errors appeared for each dataset. In particular, the average grain size was underestimated for larger grains.

Comparing this result with Figure 3, the accuracy of the training data is similar to the accuracy of the result evaluated by the number of grains, and the accuracy of the test data is similar to that of the result evaluated by the ratio of the grain boundary areas. Considering this, it is likely that the criterion for evaluating the average grain size in CNN is not the number of grains or the grain boundary area but another criterion.

After the predicted average grain size was converted into the number of grains using Equation (1), we could verify the CNN results from another perspective. A graph of the converted number of grains is shown in Figure 6. For the training data, the slope was > 0.99, and *R*^2^ was 0.999, which still showed a high accuracy. The test data showed a slope of approximately 0.97 and an *R*^2^ of 0.981, indicating a higher accuracy as shown in Figure 5, which was evaluated using the average grain area. In the case of Figure 5, the training and verification data were prepared such that the average grain size was evenly distributed within the training range, as shown in Figure 7a. As the number of grains determined from this result has a reciprocal relationship with the average grain size, it is difficult to conclude that they are evenly distributed within the training range, as shown in Figure 7b. In other words, the number of images increased as the average grain size increased, resulting in data imbalance. The prediction accuracy of the result determined by the number of grains (Figure 6b) was higher than the accuracy of the result determined by the average grain size (Figure 5a) owing to data imbalance.

### 3.3. Additional Validation of CNN

Additional validation of the CNN constructed in this study was carried out with microstructure images other than the training and validation data. The training data for network training had an average grain size in the range of 1200–2400 pixels, as described above. For further verification, the GB-type images with an average grain size in the range of 50–3200 pixels were prepared. The range of the average grain size in these validation datasets was outside the range of the training data and was used to verify the machine learning accuracy.

Figure 8 shows the results of the analysis for additional verification. In the training range of approximately 1200–2400 pixels, the same level of accuracy as for the training and validation data was achieved. This result again proves that the training and verification described in the previous section were properly performed.

In the case of a range of approximately 600–1200 pixels, which is lower than the training range, the accuracy is also high. Below 600 pixels, the predicted grain areas deviate from the correct values. This is because many small grains of less than a few pixels are buried in the grain boundary area and are not properly identified, which is not a problem in machine learning. However, when the grain area was larger than 2400 pixels, the average grain size was underestimated compared with the actual value. This phenomenon occurs when the average grain size is large, even within the training range. This error is an extension of the underestimation within the training range. Considering these results, when satisfactory regression is achieved within the training range, the network function may be universally used not only for interpolation within the training range but also for extrapolation outside this range.

Verification with experimental images and simulated CL-type images is shown in Figure 9. The prediction of the average grain size of the experimental GB-type images showed reasonable accuracy, while the CNN did not give proper estimation for experimental and simulated CL-type images across the entire range.

### 3.4. Analysis of Mid-Layer Images

When an input image passes through hidden layers, it is difficult to determine the processes involved in image processing. To better understand these processes, mid-layer images were generated after each CP layer. A simulated GB-type image, as shown in Figure 10a, was entered as an input into the trained system, and mid-layer images were generated after each CP layer. In a mid-layer image, the digit of each pixel may be out of the range of the grayscale image (0–255) during the convolution process. For a visualization of the image pixel information, each pixel was standardized with the average value and standard deviation of all pixels, and the range was adjusted to have a value between 0 and 255. The mid-layer image observed by the user does not accurately envision the hidden layer but may show the approximate conversion process.

The mid-layer images after the CP1 layer are shown in Figure 10b. CP1 received the image with three channels (RGB), underwent a convolution process with four filters, stored the result in four channels, and then generated four mid-layer images through activation and maximum pooling. The size of the image was reduced from 256 × 256 to 128 × 128 pixels. In the input image of Figure 10a, the grains and grain boundaries were designated by white and black, respectively; however, the color of the grains changed to dark gray in the mid-layer image after CP1. Except for the second image, the grain boundaries are highlighted with differing levels of brightness according to direction. This trend became more pronounced in the mid-layer image after CP2, as shown in Figure 10c. Because there were 16 channels in CP2, 16 mid-layer images were generated. The size of the image was reduced from 128 × 128 to 64 × 64 pixels. The grain-boundary component is highlighted according to the direction.

The mid-layer image after CP3 is shown in Figure 10d and appears to be significantly different from that after CP1 or CP2. The highlighted grain boundaries with respect to the direction disappeared. This is because the image size was reduced to 32 × 32 pixels and could now be stored in the form of a line segment. It appears that the image was recorded in the form of dots according to the direction and curvature of the grain boundary. For example, the image was maintained in the form of grains, but compared with the input image in Figure 10a, the number of grains was greatly reduced, and it is likely that the average grain size was not evaluated using the number of grains. The CP4 mid-layer image, Figure 10d, was 16 × 16 pixels in size; therefore, it was difficult to determine the differences between each image.

By analyzing the mid-layer images, we concluded that the CNN used in this study did not evaluate the average grain size based on the number of grains or the area ratio of the grain boundaries. The size of the filter used in the convolution is 3 × 3 pixels, which is too small to contain all of one grain in the initial hidden layers, that is, CP1 or CP2. The filters appear to detect only grain boundary segments and, in particular, the curvatures of the grain boundary segments. Assuming that a grain is circular, the curvature of the grain boundary is inversely proportional to its radius. As the grain radius increased, the change in the curvature of the grain boundary became insignificant, and the error in the average grain size increased when evaluated using machine learning.

Figure 11 shows the mid-layer image analysis of a simulated CL-type image, which could not be used to accurately predict the average grain size with the CNN used in this study. When observing the images after CP1 or CP2, an effort was made to obtain the color information of the grains. In addition, an attempt to detect the components of the grain boundary was found, but it was unclear compared with the case of GB-type images. A CNN trained with images which clearly show grain boundaries, such as GB-type images, cannot recognize CL-type images in which grain boundaries are defined by color variations of grains.

When machine learning is performed with an image in only one type of representation, it is expected that there is a possibility that the grain structure in different types or representations cannot be evaluated properly. For example, grain boundaries may have different widths depending on chemical etching. There are also grains that are elongated in some directions and/or include second-phase particles. For machine learning, which evaluates the average grain sizes of various microstructures, a database of various types of microstructures is required.

## 4. Conclusions

In this study, the average grain size was evaluated from microstructure images using machine learning. Due to a direct verification of the average grain size from the images, this method is suitable to confirm the validity of the quantitative analysis of a microstructure, unlike chemical composition or mechanical properties. A simple convolutional neural network was constructed using microstructure images, which were generated using the phase-field model. Subsequently, a quantitative evaluation of the average grain size was performed, and the following conclusions were obtained:(1)The average grain sizes predicted by the CNN within the training range coincided with the measured values with high accuracy;(2)If machine learning yields appropriate results within the training range, the accuracy of the machine learning results outside the training range is expected to be very high. Thus, the trained function may be used universally, regardless of the average grain size in the image;(3)The mid-layer image analysis shows that the CNN used in this study does not recognize the shape of an entire grain but mainly detects components of the grain boundary. In this study, machine learning was optimized in the form of a neural network that detects the curvature of grain boundaries and correlates it with the overall average grain size;(4)To apply the results of this study to actual cases, it is necessary to construct a large database of microstructures with various types of grain structures.

## Figures and Tables

**Figure 1 materials-15-06954-f001:**
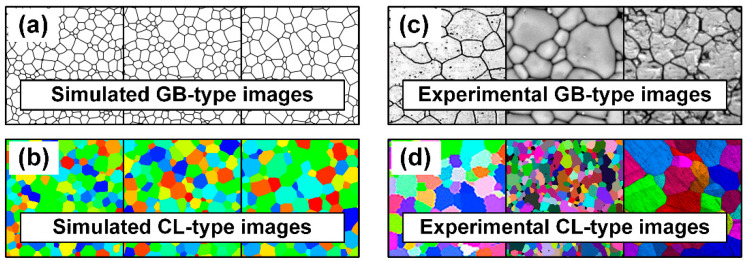
Examples of (**a**) GB- and (**b**) CL-type grains structures simulated with the phase-field model, and (**c**) GB- and (**d**) CL-type experimental images collected across the internet.

**Figure 2 materials-15-06954-f002:**
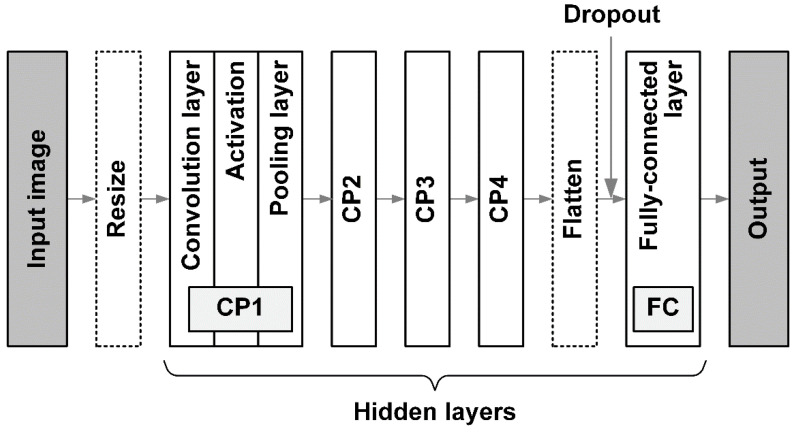
Schematic structure of convolutional neural network used in this study. Each CP layer has a convolution layer, activation function, and pooling layer.

**Figure 3 materials-15-06954-f003:**
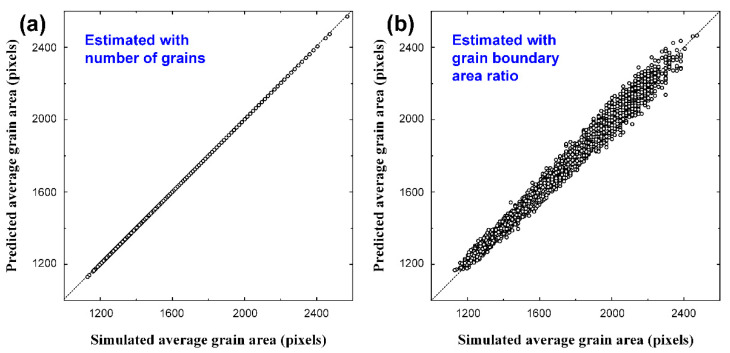
Average grain sizes estimated with (**a**) number of grains, and (**b**) ratio of grain boundary area in the simulated grains, based on Equations (1) and (2), respectively.

**Figure 4 materials-15-06954-f004:**
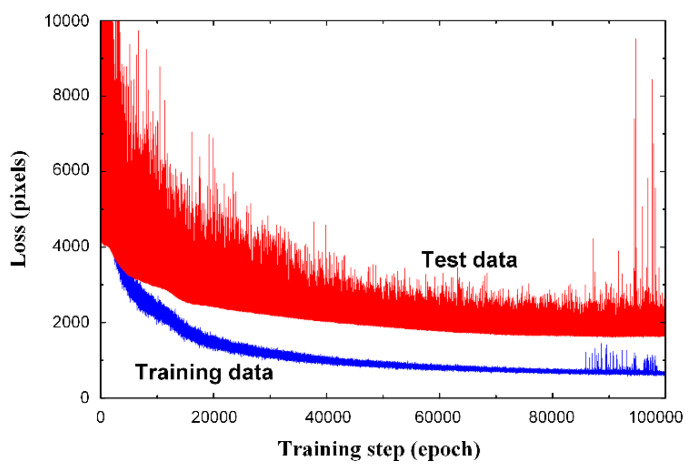
Graph of loss for training and test data during the network training. No overfitting was found up to 80,000 epochs, and high peaks of noise appeared after that. Therefore, the optimum training steps for CNN was determined as 80,000 epochs.

**Figure 5 materials-15-06954-f005:**
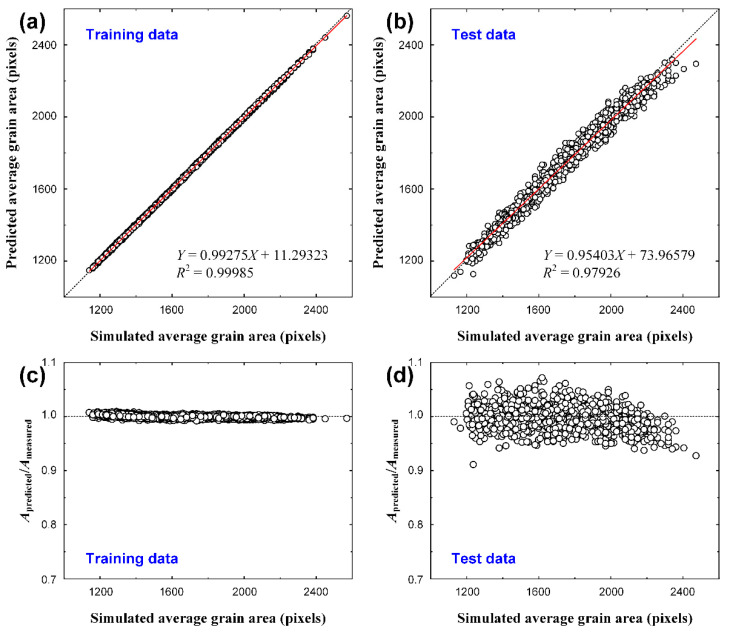
Comparison between the measured and predicted average grain areas for (**a**) 3000 training and (**b**) 1000 test data, and the comparison between the measured and predicted numbers of grains for (**c**) training and (**d**) test data. *X* and *Y* axes represent measured and predicted values, respectively.

**Figure 6 materials-15-06954-f006:**
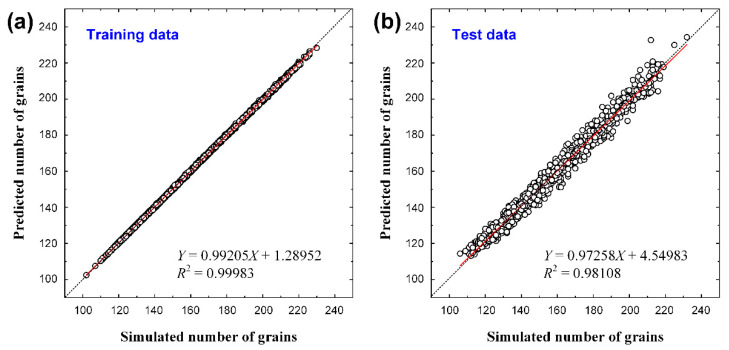
Comparison between the measured and predicted numbers of grains for (**a**) 3000 training data and (**b**) 1000 test data. *X* and *Y* axes represent measured and predicted values, respectively.

**Figure 7 materials-15-06954-f007:**
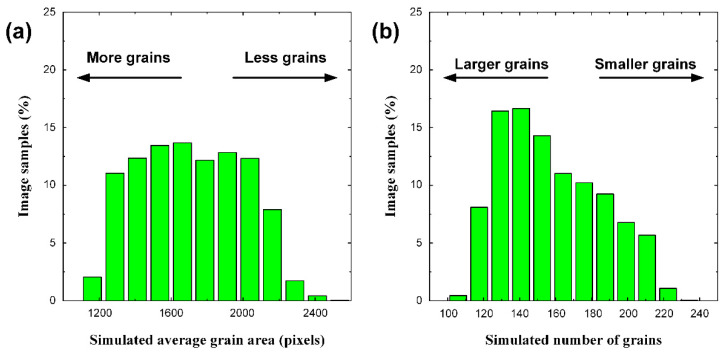
Distribution of image samples used for CNN training and test in terms of (**a**) average grain area and (**b**) number of grains. Note that the distribution for average grain area is relatively uniform, while the distribution for number of grains shows imbalance.

**Figure 8 materials-15-06954-f008:**
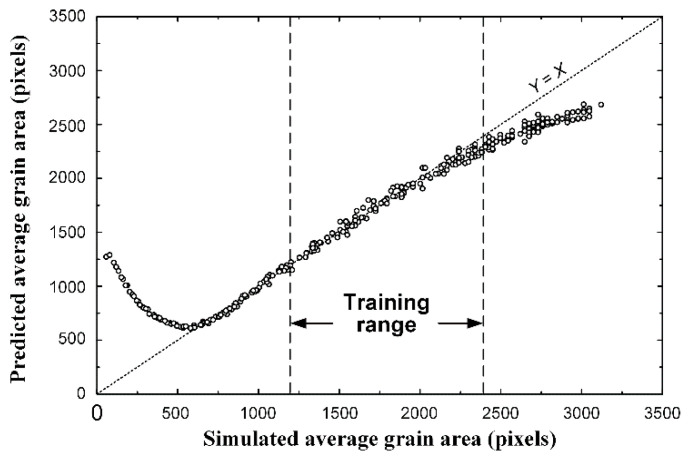
Comparison between the measured and predicted average grain area for another test dataset of 300 simulated images. In this case, the grain size ranges were between 50 and 3200 pixels, while the ranges for the training dataset was between 1200 and 2400, approximately.

**Figure 9 materials-15-06954-f009:**
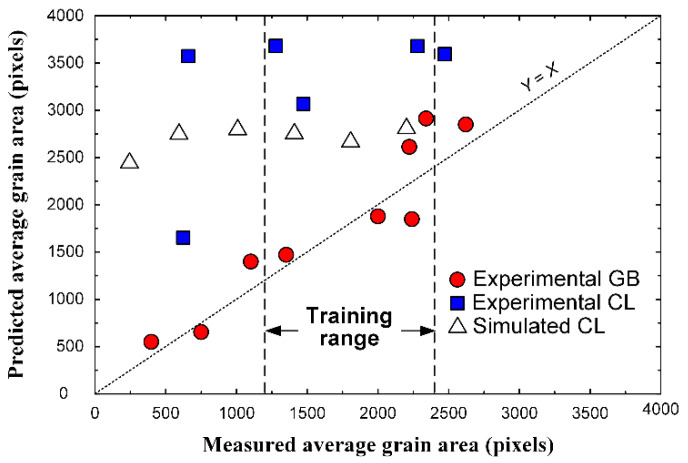
Comparison between the measured and predicted average grain area for datasets of experimental and simulated images.

**Figure 10 materials-15-06954-f010:**
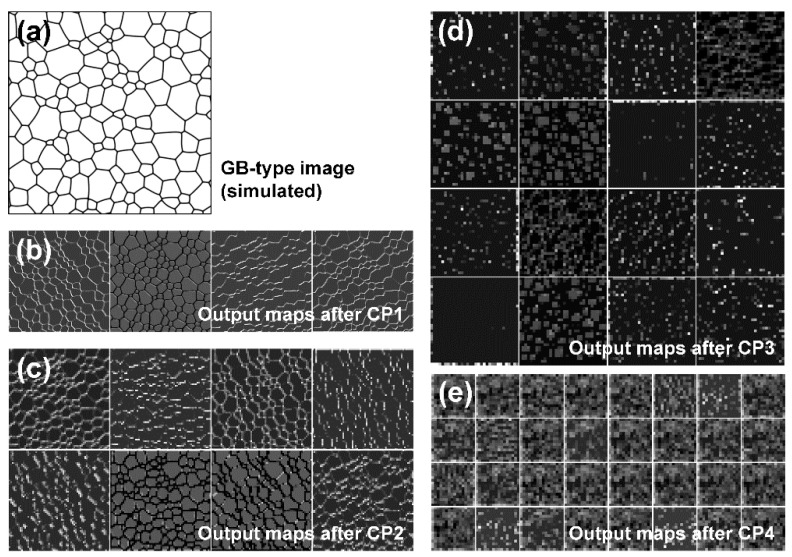
Mid-layer images for a simulated GB-type image. (**a**) Original image (512 × 512 pixels) and mid-layer images after (**b**) CP1 (128 × 128 pixels each), (**c**) CP2 (64 × 64 pixels each), (**d**) CP3 (32 × 32 pixels each), and (**e**) CP4 (16 × 16 pixels each) layers.

**Figure 11 materials-15-06954-f011:**
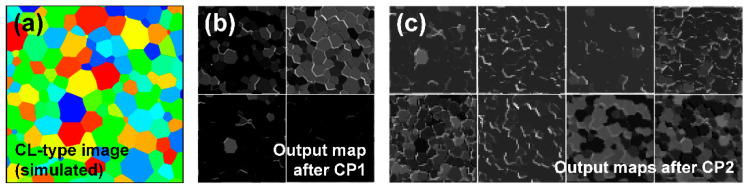
Mid-layer images for a simulated CL-type image. (**a**) Original image (512 × 512 pixels) and mid-layer images after (**b**) CP1 (128 × 128 pixels each) and (**c**) CP2 (64 × 64 pixels each) layers.

**Table 1 materials-15-06954-t001:** Model of CNN for estimation of average grain size used in this study.

Layer.	Sublayer	Input Shape	Channels /Nodes	Filter	Padding	Stride	Activation	Output Shape
**Input**	–	–	3	–	–	–	–	512 × 512 × 3
**Resize**	–	512 × 512 × 3	3	–	–	–	–	256 × 256 × 3
**CP1**	Convolution	256 × 256 × 3	4	3 × 3	1 × 1	1 × 1	ReLU	256 × 256 × 4
	Max pooling	256 × 256 × 4	–	2 × 2	0 × 0	2 × 2	–	128 × 128 × 4
**CP2**	Convolution	128 × 128 × 4	8	3 × 3	1 × 1	1 × 1	ReLU	128 × 128 × 8
	Max pooling	128 × 128 × 8	–	2 × 2	0 × 0	2 × 2	–	64 × 64 × 8
**CP3**	Convolution	64 × 64 × 8	16	3 × 3	1 × 1	1 × 1	ReLU	64 × 64 × 16
	Max pooling	64 × 64 × 16	–	2 × 2	0 × 0	2 × 2	–	32 × 32 × 16
**CP4**	Convolution	32 × 32 × 16	32	3 × 3	1 × 1	1 × 1	ReLU	32 × 32 × 32
	Max pooling	32 × 32 × 32	–	2 × 2	0 × 0	2 × 2	–	16 × 16 × 32
**Flatten**	–	16 × 16 × 32	8192	–	–	–	–	8192 × 1
**Dropout**	(0.5)	8192 × 1	8192	–	–	–	–	8192 × 1
**FC**	–	8192 × 1	128	–	–	–	ReLU	1
**Output**	–	–	1	–	–	–	Linear	Regressed value

## Data Availability

Not applicable.

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
