# Peer review of "Estimation of Average Grain Size from Microstructure Image Using a Convolutional Neural Network"

_materials, 2022, doi:10.3390/ma15196954_

Round 1

Reviewer 1 Report

I have not revisions to require, the manuscript is clear, well-structured and linear. The manuscript focus, i.e. "machine learning" for grains analysis, is well described even for not expert in the field. Two comments: 1) as written, it remains a manuscript devoted to a computer methodology: this is valuable but applications to real images (whatever form different microscopical techniques) might increase scientific interest and readers; 2) share the software could help to mitigate weaknesses described in point (1).

Author Response

Thank you for the kind review.

1) We are also sorry not to have many real images to test. Various types of grain structures, such as elongated grains and second phase particles, are planned to examine in near future. Before this test, we should improve the accuracy of the present machine learning procedures.

2) We are not ready to share our house code. To my embarrassment, the code is not so clean to show people. But, the code is very simple and plain, you may find similar CNN code anywhere over the internet.

Thank you very much.

Reviewer 2 Report

1.  Grain size should be expressed with unit

2. References in the manuscript is not in correct manner

3. Good work 

Author Response

Thank you for the kind review.

1) It was tricky to define grain size in this study. Surely, we know that micrographs usually have a scale bar. But it had no meaning to indicate the scale or unit in the grain images simulated with a phase field model. Then we decided to define grain size as the number of pixels in the grain area of a 512x512 pixel image. This is clearly stated in the text in Section 2.2. "The size of a grain was defined as the number of pixels in the grain area of a 512×512 pixel image."

2) The references were managed with a software called Zotero, as it is recommended by Materials. The instructions for authors say:

"Your references may be in any style, provided that you use the consistent formatting throughout. It is essential to include author(s) name(s), journal or book title, article or chapter title (where required), year of publication, volume and issue (where appropriate) and pagination. DOI numbers (Digital Object Identifier) are not mandatory but highly encouraged. The bibliography software package EndNote, Zotero, Mendeley, Reference Managerare recommended."

If there is something needed to further change, please let us know.

Thank you again for your review.

Reviewer 3 Report

It is suggested that the author consider the following suggestions and modify the paper:

1.    The introduction of the paper, the summary of the current research status are insufficient. The current paper review is only around the accuracy of the average grain size prediction, while the work of other scholars in this field wasn’t provided. It is suggested to add relevant references in the field and illustrate the work done by other scholars in the field of machine learning to estimate the grain size of microstructure images. Finally, summarize the research status and shortcomings, and explain the novelty and necessity of this research.

2.    In paper 2.1, “For the activation function, the rectified linear unit (ReLU) function was applied to the CP and FC layers, and no activation function was used in the output layer.” Why does this paper use ReLu function as activation function? In the output layer, why was no activation function used? Please give the relevant reasons briefly.

3.    In paper 2.1 and 2.2, according to the general process of machine learning, it is necessary to acquire the data set (microscopic images) first, and then build and train the model. However, the paper does not follow this logic. It is suggested that authors exchange the order of paper 2.1 and 2.2.

4.    In paper 3.1, “it is likely that the criterion for evaluating the average grain size in CNN is not the number of grains or the grain boundary area but another criterion.” The point is lack of sufficient explanation, although some results are provided in Figs. 3 and 5. Authors need to explain the simulation results in detail and clarify or extrapolate the “another criterion”.

Author Response

Thank you for the kind review.

  1. We added some sentences and references in Introduction:

" There have been researches on grain size measurement by edge detection of grain boundaries using digital image processing[15,16]. Moreover, grain size prediction has been attempted using image classification based on convolutional neural network (CNN)[17,18]."

  1. We changed/added some sentences:

"For the activation functions, the rectified linear unit (ReLU) function was applied to the CP and FC layers. ReLU is simpler than a sigmoid function, and results in faster training. No activation function was used in the output layer, since the regressed results do not require an activation function for classification."

  1. We changed the order of Sections 2.1 and 2.2 as you suggested.

  1. The purpose of this paper is to find the "another criterion" for evaluating the average grain size, because the number of grain or the grain boundary area do not explain well the results of the CNN machine learning. This is why we tried to extract mid-layer images from the network. We concluded that detection of curvature of the grain boundaries is the criterion for grain size prediction, as described in Section 3.4.

Thank you again for your review.